# Teaching Research Methods Courses in Education: Towards a Research-Based Culture

**João Filipe Matos [1], André Freitas [1],\*, Elsa Estrela [1], Carla Galego [1] and João Piedade [2]**

[1] CeiED—Interdisciplinary Research Centre for Education and Development, Lusofona University, 1700-284 Lisboa, Portugal; joao.matos@ulusofona.pt (J.F.M.); elsaestrela@gmail.com (E.E.); carla.galego@ulusofona.pt (C.G.)

[2] UIDEF—Education and Training Research and Development Research Unit, Institute of Education, Lisboa University, 1649-013 Lisboa, Portugal; jmpiedade@ie.ulisboa.pt

\* Correspondence: andre.freitas@ulusofona.pt

**Abstract:** This paper aims to identify and discuss what constitutes a research-based pedagogical culture in teaching research methods courses in master's and doctoral programs in education. The discussion draws on empirical results of a set of five focus group interviews with teachers responsible for teaching research methods courses in educational programs in higher education institutions in Portugal. Data are analyzed and interpreted within a situated learning framework that acknowledges the relevance of creating social learning spaces which accommodate the variety of students. The article closes with a characterization of what could be a research-based pedagogical culture in teaching research methods in advanced studies in education in the near future.

**Keywords:** research methodologies in education; teaching research methods; student-centered learning approach; pedagogical culture of research methods

## 1. Introduction

There has been a shift across most countries towards de-centralized decision making in education, giving more responsibility and mandating power to local authorities. Given greater information, less quality control, a more informed public, and a greater diversity of policy makers, the role of research for evidence-informed policy in education becomes newly important. In Portugal, research in education is mainly carried out by academics in higher education institutions (HEI) where research methods courses are included in the study plans in a variety of master's and doctoral programs (Matos et al. 2023).

It is non-controversial that solid preparation in research methods provides important knowledge and skills to undertake better research and thus significantly contribute to the educational community. Quality teaching in research methods requires developing a clear understanding of the complex relationships between the explicit syllabus guidelines of the courses, the previous competences of students, and the pedagogical options (Matos et al. 2023). It is a great challenge to teach research methods in education as the target population of students usually come with different forms of prior knowledge and is made up of diverse backgrounds, interests and expectations. The literature on teaching and learning research methodologies in education reflects several controversies regarding students' methodological understandings of research and the pedagogical challenges experienced by teachers (Nind et al. 2019; Ross and Call-Cummings 2020). Hence, it is possible to organize three main topics regarding knowledge dispositions and methodological competencies: anxiety and 'fear' of methodologies (mainly quantitative) experienced by students (Lovekamp et al. 2017; Saeed and Qunayeer 2021); the complexity of epistemological understanding and its applicability in life (Coronel Llamas and Boza 2011; Ivankova and Clark 2018); the lack of specific training on methodologies and a variety of ideological conceptions of research held by teachers (Kucukaydin and Gokbulut 2020; Talbott and Lee 2020). In this context, the type

of pedagogic research environment created is composed of mismatches between teaching and learning research methodologies in education that add complexity to a methodological scenario that is already challenging in itself. The same literature reveals scant empirical, epistemological and methodological reflection of teaching research methodologies in education (Wagner et al. 2019), indicative of the need to identify and understand a pedagogical culture in that domain (Lewthwaite and Nind 2016). From pitfalls to trends, the literature review reveals some insights for this shift to happen. The creation of pedagogical research communities among teachers and students (Wagner et al. 2019) emerges as the main topic with four characteristics: engaging students with ongoing research projects and real data (Engbers 2016; Müller et al. 2020); recognizing current research perspectives (Gray et al. 2015; Fonseca and Segatto 2021); promoting scientific autonomy, the acquisition of scientific writing skills and their dissemination (Müller et al. 2020; Motjolopane 2021); and providing active experience, partnerships and disciplinary work combined with information gathering and resource processing (Onwuegbuzie et al. 2009; Ananth and Maistry 2020). This complex learning scenario (Pedro et al. 2019; Kukulska-Hulme et al. 2022) of scientific engagement consolidates a research-based approach and debate about what makes quality research methods courses. The Research Methods in Advanced Studies in Education (ReMASE) project emerges from this evidence.

## 2. Research Methods in Advanced Studies in Education

The ReMASE project pursues the idea that teachers in HEIs will benefit from the use of a framework as a tool to design and implement research methods courses in educational programs. The project takes the idea that the quality of research in education impacts the quality of its results and therefore provides evidence that may inform decision makers and other stakeholders in education. This leads to rethinking the design of research methods courses and the pedagogical approaches taken. The aim of the project is to identify and provide research-based principles and guidelines for the design of research methods courses in education, that will be put together as a framework. The research team constituted experienced researchers teaching research methods in advanced programs in education, as well as young researchers, embracing the task of interrogating and improving the design and implementation of research methods courses. The key research question of the project is as follows: what principles and guidelines are appropriate to constitute a framework for the design of research methods courses in advanced studies in education in Portugal?

The ReMASE project is organized into three phases: (i) Phase I, which is concerned with mapping the field (theoretical and empirical)—meaning conducting literature reviews and consulting a database with all courses in Portugal; (ii) Phase II, which is the time for data collection and analysis (survey questionnaire followed by focus-group interviews) carried out by teachers responsible or involved in teaching research methodologies in education; and (iii) Phase III, which takes the results of the theoretical mapping and the empirical results to produce a framework—constituted by principles and corresponding guidelines—for the design of research methods courses (Matos et al. 2023).

This article aims to discuss the relevance of a pedagogical research-based culture in teaching research methods in master's and doctoral programs in education. The discussion draws on empirical results of a set of five focus group interviews with teachers responsible for teaching research methods in education in HEIs in Portugal. Given the crucial role of a conceptual framework (Eisenhart 1991) in producing evidence, the results are interpreted within a situated learning framework (drawing upon the work of Lave and Wenger 1991 and Wenger 1998) that acknowledges the relevance of creating social learning spaces which accommodate the variety of students. The discussion closes with a characterization of what could be a research-based pedagogical culture in teaching research methods in advanced studies in education in the near future.

The following key research question is addressed: what constitutes a research-based pedagogical culture in teaching research methods courses in advanced studies in education?

### 3. Methodological Approach

Within the ReMASE project, the general research problem was addressed through the (i) a content analysis of the Research Methods in Education (RME) syllabus courses' description (objectives, learning objectives, content programs, teaching methods, learning activities, assessment and mandatory bibliography) that structure the RME syllabus courses, with (ii) the empirical data coming from a survey questionnaire given to teachers responsible in those courses, and (iii) a set of focus group interviews with some participants from the previous phase of data collection (Matos et al. 2023).

Concerning the RME courses in Portugal, we identified 214 master's and doctoral programs, accredited by the national responsible agency, operating in the 2021/22 academic year, with available course syllabus information. The 214 programs are distributed across 195 master's programs in education (N = 86) and in initial teacher education (N = 109), and across 19 doctoral programs in education (N = 15) and in teaching (N = 4). Most of the educational programs in HEI offer some type of research methodologies courses or modules. Just a small part (less than 10%) of the programs set as optional the research methods courses. However, there are programs that do not include any type of course in research methodologies. Despite the type of institutional options, the main areas that are included in the research methods courses are specific competences; transversal competences; authorial/ original work; and applicability and transferability.

In this article, for the purpose of relevant discussion of research-based pedagogical culture creation, the report is centered on data coming from the focus group interviews.

The focus group interviews were conducted with 20 teachers organized into 7 groups. The selection criteria combined the rule that every focus group should have 3 to 4 teachers from different institutions plus one moderator chosen among the ReMASE project members. Each focus group included teachers responsible for RME course teaching from different Portuguese HEIs (private and public universities and polytechnic institutes). They were selected from those people who previously answered the ReMASE survey questionnaire and who volunteered to be interviewed by the research team. An invitation was included at the end of the survey questionnaire to allow participants to voluntarily participate in a follow-up focus group interview. This sampling criteria were defined by the research team considering the possibility that those who volunteered were most probably the ones who were more eager to share their experience and perspectives on teaching and learning RME.

To save time and traveling costs, the focus group interviews took place through the Zoom video conferencing system (provided by the ReMASE project's host institution). The interviews were conducted by one of the research team members who had the support of a second researcher. The guidelines and protocol previously defined were applied and the interview's average duration was 90 min.

Informed consent was obtained from all teachers involved in the study. The interviews were videorecorded with permission from all the participants in each focus group who signed an informed consent form acknowledging the purpose of the data collection, the procedures to preserve confidentiality and the actions implemented to protect the participants' identity. All appropriate ethical issues were addressed and the correspondent procedures were previously approved by the Ethics Committee of Interdisciplinary Research Centre for Education and Development-Lusófona University (the ReMASE project's host institution). The study was conducted in accordance with the Declaration of Helsinki.

The construction of the content guidelines for the focus-group interviews and protocol of its application were developed in two phases; firstly, a draft was prepared considering the project's key research questions and the trends identified in the results of the survey questionnaire given to teachers. The guidelines and protocol of application were reviewed and adjusted by other three team members, resulting a final version for implementation (see Appendix A). The protocol for implementation describes the strategies, timing, phases to contact the selected participants for interviews and the researcher guidelines.

The focus of the interviews was placed on the following key items: the relative degree of importance of the background methodologies of technical research methods; time

allocated to the various themes included in the Research Methods in Education (RME) syllabus course; pedagogies used in teaching; the degree of involvement of students in hands-on research activities (data analysis, assessment of articles, production of essays and literature reviews, etc.).

The video recordings of the focus group interviews were uploaded to and organized in the NVivo software to create the conditions for (a) a systematic and recurrent analysis to be carried out by a group of researchers, and (b) an organized review of the analysis from other members of the project's research team. In the first step, a priori categories were used following the concepts derived from the theoretical mapping of the field (Matos et al. 2023). In particular, we concentrated on two main categories: the objectives of the RME syllabus courses and their learning outcomes.

For the purpose of identifying, in teachers' voice, traces of a research-based pedagogical culture in teaching research methods, data were additionally analyzed using a situated learning-based conceptual framework (drawing on the work of Wenger 1998) that includes the following categories: (i) engagement in activity, that unfolds in forms of engagement, mutuality processes (interaction facilitates the definition of joint tasks), competence (taking initiative in the social space and creating conditions to make it knowledgeable to others, creating accountability instances), and the continuity of practices (through producing reificative memory as well as participative memory); (ii) imagination which includes the subcategories of orientation (creating possibilities for people to realize their position in the learning space, creating possibilities for people to locate themselves in time, creating possibilities for people to find themselves located in the shared meanings, creating conditions for people to locate themselves in the power relations within the class and the scientific community in general), encouraging reflection (acknowledging consciousness of learning and developing (the self and community), and promoting exploration (acting through initiative and intentionality in participation) and sharing (intentionally showing difficulties and successes); (iii) alignment that includes both convergence (participants sharing a common telos) and coordination (participants intentionally finding points of articulation and ordering them). Instances of these processes were identified in the resources and the artifacts used by the participants (teachers and students) as described by the teachers in the focus group interviews.

Those categories were used in a dialogic process for the generation of meanings and sub-categories following a grounded-theory-based strategy; the research team tagged ideas and concepts and succinctly summarized them, the codes being grouped into higher-level concepts, and then into categories. Therefore, the strategy for the analysis followed a top–down (from a priori categories to data) and bottom–up (from data to theory-based categories) double movement creating evidence for the specification of a research-based pedagogical culture that takes a situated learning stance. The focus group interviews' transcripts are anonymous, respecting the ethical framework referred to above. In this text, the participants are identified with a number (e.g., P4) after reference to the focus group's code (e.g., FG1); for example, FG1-P4 indicates teacher P4 of focus group 1. Data analysis for this paper involved two members of the research team for around 80 working hours.

## 4. Results: The Research Answers to Pedagogic Culture of Teaching RME

The literature review (Matos et al. 2023) shows pitfalls and trends (regarding practices, interactions, commitments, and knowledge) in the teaching of research methods in education. Regarding the pitfalls of the pedagogic environment, the situation consists of a practice of isolation and loneliness, disagreements and misunderstandings, commitments of discouragement and disaffection, and a type of knowledge that is disconnected and fragmented. Despite this fragile picture, the literature makes it possible to reveal other possibilities of teaching research methodologies in education. In this respect, trends can be organized into the following: a practice formulated by collaborative ways; an interaction organized through active pedagogies where teachers can act as learners and learners as experts; commitments that are centered on the elaboration of a scientific culture promoted

by teachers in which autonomy is increased in students, and in this idea the knowledge organized is student-centered, promoted through hands-on course designs.

### 4.1. Purpose and Learning Outcomes in RME Courses

The production of a RME course follow a process based on the identification of its constitutive elements (objectives, learning outcomes, contents, working methods, and forms of assessment) in similar courses in other HEIs. The main strategy for elaborating on the course plan relies on the contents to be delivered afterwards, combining and articulating them with objectives and learning outcomes. This may be due to a lack of background in research methods and an absence of a more general view of the purpose of the master's or doctoral program in which the course appears. However, it is possible that a perception of the pressure inherently perceived by the institution responsible for the formal permission to run the program (the national Agency for Assessment and Accreditation of Higher Education) may have an important role in that matter. This is illustrated in a teacher's statements:

> "what I usually do when available is to go and check at other universities what are the contents that are usually covered in that course, not so much to copy, but to try to have something common to the other universities so that more or less the same contents or the same materials so that later there is no difference and there is not always to compare (to other major universities) ( . . . )" (FG1-P4).

> "( . . . ) Additionally, it is from there that I make the curricular plan of the course, the objectives, the methodologies, the contents, the teaching strategies and then the assessment that will have to be coherent with everything that is behind as well" (FG1-P4).

A serious implication of this largely adopted process of producing a course is that a sort of equalitarian trend is visible in most of the RME syllabus course's description. For example, referring to the objectives of the RME syllabus course, more that 80% of the course description included in master's programs use formulations such as "taking into account the skills that are intended to be developed in this master's degree, the purpose of this course is to carry out an introductory approach to the most important principles of scientific procedure, the stages of its practical application and the main methods and techniques of research in education".

However, teachers recognize that a content-based organization of a RME course evades much of the purpose of such a course as illustrated by teacher FG4-P3:

> "( . . . ) students sometimes have difficulties, therefore, in formulating research objectives and questions, even because they confuse objects, research objectives with teaching-learning objectives ( . . . ), we are always in the struggle . . . and therefore this is, perhaps, our biggest problem" (FG4-P3).

In addition to the discomfort shown by the teacher, it is apparent the lack of clear view of the purpose of a RME course in relation to its audience:

> "(it is important) to learn to know the structure and importance of scientific thinking. Hence, in undergraduate studies you have to understand the structure, (it is) an introduction to scientific thinking. It means that it is a phase, it is a level in which you must know the importance, relevance, and the form of its structure, reading and analyzing some scientific articles. A master's degree, in contrast, in the Bologna model, is already a level where research is taught ( . . . ). That's right, but the master's also has different audiences, different objectives, some of them are professional and the construction of thinking is not always relevant to their curriculum. Hence, we will have to take a good look at what is the audience, what is the objective of the master's degree so that we can really build a curriculum that leads and teaches how to build scientific thinking" (FG4-P3).

In general, the RME syllabus' course description include as important learning outcomes the identification and use of different types of methodological strategies, mastering their theoretical, methodological and technical requirements, so that they can make appropriate choices. Additionally, teachers generally agree that "understanding scientific rationale" should be included in every RME course's learning outcomes. However, while teachers underline the relevance of understanding how science is produced, they quickly jump to the list of topics to be addressed (such as research paradigms, types of research studies, emphasizing qualitative versus quantitative approaches, and methods for data collection), showing difficulties in the articulation of so called "theoretical items" with practicalities for data collection.

*4.2. Content-Centered Pedagogies*

As suggested in the section above, teachers value the efforts put into helping students understand elements of the theoretical background of research in education although this is a difficult task both because of students' lack of previous knowledge as well as because the syllabus's course specific contents are supposed to be covered in classes.

> "( . . . ) at least from the experience I have, ( . . . ) the most critical part is in data collection and analysis . . . do a 15-min interview and treat the interview, make a small questionnaire, etc . . . but creating the instruments because a questionnaire is not a list of questions, an interview is not a list of questions and, therefore, from the idea of how it is done to how it is applied, of course, (you have to deliver) a minimum on how you can analyze quantitatively, qualitatively, a simple descriptive quantitative" (FG2-P2).

It is apparent that most teachers believe that concrete methods for data collection are at the kernel of the issue of teaching research methods; therefore, they center their working methodologies with students around successful strategies for that purpose. This has an impact on students' autonomy and ability to methodologically address concrete research problems, as demonstrated below:

> "( . . . ) as an advisor I see that they have a lot of difficulties, they usually have a lot of difficulties in operationalizing the theoretical concepts that we are dealing with (in RME courses)" (FG2-P1).

Teachers make extensive references to pedagogies that include project-based learning and problem-based learning, a flipped classroom and collaborative work although they tend to be in full control of the class activities and assume the initiative for all those activities. The syllabus course (in particular, the contents to be covered) tend to play a strong structuring role as a resource that teachers use and that formats their own pedagogies. These are circumstances that clash with the complexity of understanding RME beyond abstract conceptions. The lack of application of knowledge, in relation to this range of pedagogies, creates constraints on teaching and learning RME courses.

Data collected within project ReMASE show that in the doctoral and master's RME syllabus courses' description, there is a prevalent (around 90%) mention of "theoretical and practical teaching" although only 10% refer to "practical laboratorial teaching". Most of the RME courses in doctoral programs refer to "seminar" as the working model although it is not clear how seminars are conceptualized or the role of students and teachers in those activities. It becomes apparent that the notion of a seminar is connected more to doctoral programs than to master's programs but there is no explicit reference to the responsibility of students in those seminars. In fact, in some cases it comes up that seminars are more a way to describe classroom activities based on sharing and discussion, therefore not having a clear seminar-like identity. Teacher P1 in focus group interview FG-5 addressed the way seminars operate:

> "Then, in the seminar, we respond to specific problems, for example, ( . . . ) when we are building data collection instruments, in these classes those who make questionnaires or who make interview guidelines, they show them to their

colleagues, the seminar teachers and usually with the methodology teacher, ( . . . ) and then everyone analyzes and suggests changes. This is important because sometimes colleagues put themselves in the respondent's shoes and remember that there is still one more answer option missing, that those answer options are not mutually exclusive ( . . . )" (FG5-P1).

At the same time, that teacher referred to extending the discussion on how appropriate it is to use a given instrument to address a particular research problem and brought in the issue of time dedicated to that discussion. The time allocated to the analysis and discussion of research approaches and methodologies is referred to by all teachers as a key issue in teaching RME courses. However, this brings in the traditional structural separation between RME courses and other courses of the study plan that represent the current model of organizing master's and doctoral programs in education in Portugal.

In summary, we can point as a key issue to the fact that teachers focused on the research methods contents to be taught and easily set aside the purpose and learning outcomes that are formulated in the design of the courses. When a teaching curriculum supplies structuring resources for learning and controls access to it, the meaning of what is learned is mediated by an external view of what knowing is about (Lave and Wenger 1991).

*4.3. Teachers' Role and Responsibility*

If teachers show a sense of discomfort regarding the way they implement the RME courses—claiming for more time and more opportunities to go deeper in important dimensions of research methods, and complaining about students' previous knowledge—the fact is that they manifest a concern with their own preparation to teach in the area. All teachers interviewed said that they had no specific preparation to teach RME courses apart from the practice of research during their advanced studies and, in some cases, a few short courses on research methods. However, they referred to the importance of the practice of advanced research to be able to recontextualize the knowledge that emerges from that practice into teaching RME courses. Teacher P3 interviewed in FG2 declared the following:

"( . . . ) my education (in RME) as a researcher both in terms of the training courses of the postgraduate programs I attended and in participation (in research projects) is a decisive aspect. The participation I had in research projects (provided) all aspects of nature, both theoretical and practical, which had been discussed at the level of methodology ( . . . ) and the application of its practical aspects in quite different contexts ( . . . )" (FG2-P3).

This is a very important aspect that teachers bring into the issues of teaching—the relationships between undertaking research and teaching in higher education. They all recognize this as part of their own professional development efforts but eventually feel the need for a stronger background and for the structural organization of ideas in research methods.

"(in research practice within projects) with a greater incidence in methodologies of a qualitative nature, quantitative nature, mixed nature, therefore going deeper in research projects, we went deeper into certain specific methodologies of inquiry. In addition, another work that is extremely important is individual reading, that is, that we are doing based on the various research manuals and literature, this ends up being extremely rich . . . and also because they respond to what our needs, our desires, to look specifically, to delve into certain areas in accordance with our interests and our needs" (FG2-P3).

Teachers explicitly show that their background in research methodologies comes from their individual efforts in recontextualizing their research practice into forms of addressing research methods, issues, techniques, and problems. The risk of reification of methods is certainly present in that process of recontextualizing the research practice in teaching RME courses—in fact, there were traces of that phenomenon in the voices of teachers. Teacher

P4 of FG2 provided a clue to try to overcome that risk within the process of designing the activities with students:

> "How do I bring my research center to my classes? Exactly with my examples that I can bring in and basically with examples of investigations already carried out, investigations that are ongoing, in which they participate, that the research center as such has also supported" (FG2-P4).

> "Then there is an education that I consider important, which is my experience as a researcher. This is where I understand the main difficulties and problems and the main ethical questions that arise and I also think that at this level the education is very useful when sharing it with my students" (FG4-P2).

In summary, teachers recognize their responsibility in leading activities that create conditions for students to develop appropriate knowledge and competence in research methods and in most cases bring in their own experience in research. Even if they assume that their preparation to teach RME courses is part of their efforts in professional development, a feeling of discomfort emerges when they recognize that mostly they have no specific academic preparation to teach in the area.

### 4.4. Towards a Research-Based Pedagogical Culture

Data collected in the focus group interviews show both the effort of teachers in designing and teaching RME courses as well as a discomfort partly with the results of their work and the way it is implemented. We argue for a need to support teachers in rationalizing the (implicit) need for a research-based pedagogical culture that emerges from their voices.

#### 4.4.1. Time in a Research-Based Pedagogical Culture

There are indications that teachers are aware of the need to improve the way RME courses are designed and implemented. Perhaps the most commonly referred to issue is time dedicated to learning research methods in education. Teacher P2 interviewed in FG4 was very explicit about this and linked the issue of time with the maturity of ideas when students deal with research methods:

> "The first issue is really the question of the time given for students to integrate information about scientific thinking and the way how scientific knowledge is said to be valid. I (have to) guarantee that people validate what I build. Therefore, the question of validating the scientific process is one of the questions that one must address permanently and cannot be resolved with little experience. Validating a survey requires a lot of experience and a lot of training. Otherwise, it's a joke that can't be accepted as scientific thinking. That would be the primary issue—time" (FG4-P2).

The time for learning represents an entry point to the discussion that teachers bring in on how to make proper changes to improve students' preparation on research methods. They understand that the academic organization and structure of master's and doctoral programs in Portugal add constraints that act against the prime objective of those programs: to develop high-level understanding and competence in research in education. This means that teachers are aware of the space needed for students to take initiative in RME courses and the conditions required to make that initiative knowledgeable to others, thus contributing to stimulating learning. This reference to time is both connected to the time of the student as well as the time of the teacher (to teach). With reference to the time of the student, it appears obvious that the difficulty of ambition can be explained by the structural organization of RME courses into 2 or 3 contact hours per week. Teacher P4 from FG-2 spoke of the sort of activity elected as more productive:

> "( . . . ) the process of analysis of works and research itself, therefore closely linked to the issue of articles and dissertations ( . . . ). Together with that analysis, we

make the students also plan methodological approaches in the sense that it is easier for them to apprehend the concepts they read in such guiding texts when they are confronted with that need. ( . . . ) often we are realizing and identifying (needs) for the development of the dissertation" (FG2-P4).

Although we have many references to time in teaching RME courses, teachers draw from the unamendable and rigid definition of time given by institutions to plan and work with students. In fact, time is an important part of the culture of school in general and plays the role of a structuring resource that determines most schooling activities. That is why it must be challenged by taking a critical stance on its orientational role in teaching RME courses. As participants in an academic culture, students need a form of orientation that helps them to locate themselves in the learning space and this is carried out through creating opportunities for them to locate themselves in time and find themselves located in the meanings that time takes. The time for reflecting and acknowledging the consciousness involved in learning and development is crucial. Reflection and exploration involve acting and initiative as well as intentionality in participation, therefore creating (intentionally) opportunities to show the difficulties and successes which are all together inherent to learning.

### 4.4.2. Participation and Engagement in a Research-Based Pedagogical Culture

Students' participation in research as a way of learning research methods means absorbing and being absorbed in the culture of the practice of research. If this principle—formulated from a situated learning perspective—carries potentialities for the design and implementation of RME courses, teachers face difficulties in attracting students to modes of participation that are not part of their schooling culture. This was a key idea in the voice of teacher P2 interviewed in FG-4:

"What happens, and I have not had success with, I clearly say, when students start thinking about research and, above all, when they start thinking about their research projects or their research work to present the final dissertation, they become very much in control of their research problem, and it seems that they do not want to share it" (FG4-P2).

We see in these traces of dissatisfaction and even discomfort with the lack of engagement of students in collective activity participation in the RME course. However, we understand that participation involves mutuality (Wenger 1998) that is apparent for example when joint tasks are defined by participants and interactional facilities are available. The very idea of participation in course activities involves allowing peripherality and in fact the consideration of dynamic trajectories of participation. It is not clear if teachers are conscious of the relevance of attributing responsibility and power to students regarding their own modes of participation. This is part of a research-based pedagogical culture, that necessarily includes the production of reified memories (e.g., an article or a student project presentation in class) as well as corresponding participative memories (Wenger 1998).

Teachers value the participation of students in all activities in the RME courses although they tend to assume participation as oral intervention, setting aside more engaged forms of participation. Teacher P1 of FG-5 evaded this light interpretation and went into more significant forms of participation:

"(another important issue) is student participation in research projects, but in a way that is ethically integrated. We cannot do research at the expense of our students, we cannot, that is not what it is about and that is very clear (for me), but it is very important that a student understands how to transcribe an interview and that she goes through this experience before to decide (about the methods to use)" (FG5-P1).

Learning as participation—assumed as a situated phenomenon (Lave and Wenger 1991)—is associated with the evolution and transformation of the individual's participation and their sense of belonging. Teachers realize that the notion of participation includes

simultaneously a process and a product. This would mean that students, more than being equated at the individual level, are thought of as participants in the social learning space constituting the class of RME courses.

The unit of analysis in research necessarily encompasses simultaneously the person, the activity and the contexts in which it unfolds. Additionally, this premise has implications in terms of the way teaching research methods is conceptualized, designed and implemented by teachers in master's and doctoral programs.

### 4.4.3. Shared Repertoire as Element of a Research-Based Pedagogical Culture

Research inherently has an intentional basis and a productive character, linked to the transformation of an object into a given outcome or result. The essence of research activity is the production of new structures of understanding a social activity, that are objectively oriented, creating new objects. The very object of research can be identified with productive social practices, in their diversity and complexity, thus distancing from the idea of learning as reproduction. Research is intrinsically linked to the capacity for creation and innovation.

A central element of any culture is the repertoire of its members—the participants in the practices that make up that culture. In RME courses, teachers put most of their efforts into enabling students to acquire a shared repertoire (Wenger 1998) whose visible content is apparent in students' forms of writing and talking about research, but this includes core concepts, strategies, processes, terminology, etc.

Most of the teachers interviewed reported a contradictory situation referring to the idea that on one hand students are able to describe a 'menu' of possible research approaches and methods but on the other show difficulty in making sense of the relationships between research problems, substantive theory and methods. Teacher P1 interviewed in FG-5 reported her forms of working with students in order to both avoid a menu-like research methods approach as well as to follow the norms and structure of the RNE course curriculum:

> "We give (to students) a practical situation, a research design and ask them to write as if they were building the methodology section. Because, it is not easy indeed. These are skills and abilities that require a lot of writing work. And, we also ask them to do a research project, it is not (necessarily) their individual project, but it is for them to realize that there are a set of steps (to do research) ( . . . ) in this process there is also a product that is delivered and that should follow criteria of criticism, of scientific writing, etc. And then there's a test, but the test is more about application, so, yes, they are application exercises" (FG5-P1).

In this statement, it is revealed that the teacher attempts to domesticate the processes of learning RME by combining "knowledge" of forms of planning research and the concepts and the associated terminology with a curricularization of that knowledge, postponing the practice of research. This is visible in the forms of assessment that are indicated.

The repertoire that is shared, throughout participation in the social practice in learning RME, goes far beyond the knowledge of terminology. A repertoire includes forms of organization of the practice by the practitioners and the artifacts that mediate the processes inherent to the practice. If, for example, we consider a questionnaire as an artifact that is appropriate to collect empirical data to address a certain research problem, it should be understood not as an object (external to the practice) but as part of the pedagogical culture of teaching RME courses.

### 5. Conclusion: Rationalization of Research-Based Pedagogical Culture in Teaching RME Courses

The analysis of the focus group interviews, together with the literature review of teaching RME (Matos et al. 2023), indicates a bittersweet sense of what should be considered relevant changes in teaching research methods in education in advanced studies in education. From this, we identify a mostly implicit call for a transformation of the pedagogic culture that informs teachers' practices in RME courses.

### 5.1. Constitutive Elements of a Research-Based Pedagogical Culture

A teaching culture based on research should adopt principles that value the participation of teachers in research. This would entail (i) emphasis on inquiry practices, promoting the use of inquiry-based learning approaches that encourage students to investigate problem and generate working hypotheses and forms of addressing them with the proper tools; (ii) data-driven decision making, because a research culture should encourage (in a recursive way) the use of data to inform teachers' decision making, necessarily including the use of formative assessments to monitor students' progress and adjust activities accordingly; (iii) collaboration among teachers and with other researchers to share ideas and create opportunities to learn and practice research; (iv) professional development prioritizing ongoing professional development opportunities informed by research to support teacher growth and continuous improvement; (v) student-centeredness because a pedagogical culture should place a strong emphasis on student learning, with a focus on creating learning scenarios (Pedro et al. 2019; Kukulska-Hulme et al. 2022) that are responsive to the needs and interests of diverse learners; (vi) rigor in the analysis and discussion of pieces of research informed by the relevant literature.

The key idea resulting from the findings that contribute to the state of the art in teaching and learning research methods in education concerned with creating a research-based pedagogical culture is participation. Putting participation at the kernel of a research-based culture involves creating opportunities for active engagement and involvement from all stakeholders, including students, teachers, researchers, and education community members.

#### 5.1.1. Student Participation

Student participation is crucial when we assume that learning is an integral part of the social practice (Lave and Wenger 1991) in which the class is involved. Therefore, student participation is translated into access to activities that encourage inquiry, reflection, and collaboration (Lewthwaite and Nind 2016). This can take the form of classroom discussions, group work, student-led research projects or any other kind of activity that allows students' agency to encourage their participation and engagement in situations and problems (Matos et al. 2023).

#### 5.1.2. Teacher Participation as Researcher

Teachers are inherently participants in classroom activities. However, the form of contribution to creating a pedagogical culture based on research should value their involvement in research-based professional development opportunities with their peers (e.g., within the organizational structure of a concrete collaborative research studies framework) and involve engaging in dialogue with other teachers and researchers to share best practices and lessons learned (Wagner et al. 2019).

#### 5.1.3. Educational Community Participation

Members of the local or national educational communities can be engaged in the research process through partnerships and collaborations that promote the co-creation of knowledge (Wagner et al. 2019). This may involve working with community-based organizations, parents, and other stakeholders to design and conduct research studies that address local issues and concerns.

### 5.2. Reification in a Research-Based Pedagogical Culture

Participation and reification constitute any practice (Wenger 1998). In the design and implementation of RME courses, it is required to consider as equally important the reified products of participation. In a pedagogical culture based on research, reification translates into tangible outcomes that result from the active engagement and involvement of participants in the research process.

### 5.2.1. Research Studies

Research studies are the most common reified product of participation in a pedagogical culture based on research. These studies can take various forms, including empirical studies focused on addressing educational problems or issues. This is the most commonly referred to form of tangible reification in RME courses. At stake are the epistemic, methodological and ontological specificities of the different scientific rationales in education, the ways of operationalizing educational research and even the ways of organizing and publishing the knowledge that is produced and understandable to the educational community, which, e.g., in Portuguese studies, vary from quantitative approaches assuming a descriptive and exploratory nature (Piedade and Pedro 2019), to qualitative approaches adopting narrative and poetic methodologies (Freitas et al. 2020). This is a reflection that, on an international scale, is limited by the boundaries of science itself, but with increasing epistemological, methodological and ontological breadth, and is therefore complex to understand and act upon.

### 5.2.2. Curriculum Materials as Artifacts

Participation by teachers and students in the research process can result in the development of new curriculum materials that are grounded in evidence-based practices (Saeed and Qunayeer 2021). These materials can include lesson plans, instructional strategies, and assessment tools, and are designed to improve student learning outcomes.

### 5.2.3. Professional Development Resources

The participation of teachers in ongoing professional development opportunities can result in the creation of new resources that support teacher growth and development (Fonseca and Segatto 2021). These resources can include workshops, training materials, and online courses that are based on the best practices and grounded in research.

### 5.2.4. Community Partnerships

Participation from community members in the research process can lead to the development of new partnerships and collaborations that support the co-creation of knowledge (Ivankova and Clark 2018). These partnerships can result in the development of new programs, initiatives, and policies that address local educational issues and concerns.

### 5.2.5. Dissemination Materials

Dissemination materials are products that are designed to communicate research findings to a wider audience. These can include research reports, academic articles, conference presentations, and other materials that are intended to share research findings and insights with other researchers, practitioners, and policymakers.

Perhaps the most relevant form of orientation towards the construction of a research-based pedagogical culture in RME courses refers to forms of participation in the social practice that takes place in seminars or classes. Participation in practices in which knowledge of a research method results in reified products (in fact, the domain of research practice) is an epistemological principle of learning research methods. The social structure of participating in research practice (that includes the diverse trajectories of students and teachers), together with the inherent power relations and the conditions of legitimacy, both define the possibilities of participation and, therefore, of learning.

**Author Contributions:** Conceptualization, J.F.M.; methodology, J.F.M. and A.F.; software, J.F.M.; validation, J.F.M., A.F., E.E., C.G. and J.P.; formal analysis, J.F.M.; investigation, J.F.M., A.F., E.E., C.G. and J.P.; resources, J.F.M., A.F., E.E., C.G. and J.P.; data curation, J.F.M.; writing—original draft preparation, J.F.M. and A.F.; writing—review and editing, J.F.M. and A.F.; visualization, J.F.M., A.F., E.E., C.G. and J.P.; supervision, J.F.M.; project administration, J.F.M. and E.E.; funding acquisition, J.F.M. and E.E. All authors have read and agreed to the published version of the manuscript.

**Funding:** The article's publication was financed by national funds—F.C.T. (Fundação para a Ciência e Tecnologia, I.P.), in the scope of the project EXPL/CED-EDG/1130/2021.

**Institutional Review Board Statement:** This paper was developed under the scope of a research project approved and funded by F.C.T. IP (national agency for research, science and technology) under contract # EXPL/CED-EDG/1130/2021. All ethical issues were considered and approved by the Center for Interdisciplinary Studies in Education and Development-CeiED of Universidade Lusofona, Portugal.

**Informed Consent Statement:** Informed consent was collected from the participants verbally and in writing before the focus group interviews were conducted.

**Data Availability Statement:** Data is unavailable due to privacy and ethical restrictions related to its confidentiality.

**Conflicts of Interest:** The authors declare no conflict of interest. The funders had no role in the design of the study; in the collection, analyses, or interpretation of data; in the writing of the manuscript; or in the decision to publish the results.

## Appendix A

Project ReMASE

Guidelines and protocol for focus group interviews

General objective: understanding professional practices and respective pedagogical and scientific characteristics of teachers responsible for teaching RME courses.

Invitation to the FG interview was sent in advance briefly explaining the orientation of the study and its objectives and explaining the orientation of the focus group and its objectives.

Introduction: introduction of participants by team members. Scenario: online interview via Zoom videoconferencing tool.

Context and transition: invitation of participants to share their pedagogical experiences in teaching research methodologies in education.

Permission in writing for the videorecording of the focus group interview was obtained at the time of scheduling it. Permission was orally confirmed at the beginning of the interview.

Key Discussion 1—prior methodological knowledge.

A. Pertinence and characteristics: Research methodologies in education make a substantial contribution to student training; should this contribution be more in the practical or theoretical domain? What is the relevance/what is most characteristic and distinctive about the course for student learning in your specific program?

B. Attitudes and representations: Do students come prepared from the master or doctoral program with the expected level of understanding/interpretation that is required?

C. Are faculty qualified to teach research methodologies? How are they trained?

Key discussion 2—pedagogical and scientific practice.

D. Work/teaching methods and activities: What organizational and pedagogical orientations, communication styles, technical devices, resources and activities can be identified in your practice? How do you approach research and research methodologies while teaching technical/operational skills?

E. Recommendations: What are the best pedagogical and scientific practices? What scientific processes can be adopted to promote hands-on/hands-on learning?

Key discussion 3—pedagogical and scientific culture.

F. Attitudes and representations: What is the relationship between research concepts (in their different dimensions) and the culture built between students and professors, namely through the RME course?

G. Recommendations: What pedagogical processes should be adopted to promote an environment of inquiry?

Key discussion 4—institutional environment.

H. Scientific management: How does the institution perceive research impact on the educational offer of RME courses?

I. Physical, material and human resources: How can higher education institutions ensure that the objectives of teaching quality research methods in education courses will be achieved?

Key discussion 5—being a teacher-researcher and student-researcher

J. Attitudes and representations: Is research a profession per se or is it an activity implied in a profession? Are students, future teachers, researchers in training?

K. Recommendations: What pedagogical and scientific processes can be adopted to engage students in research methodologies in education? What possibilities do you see for the future?

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
