# Peer review of "Teaching Research Methods Courses in Education: Towards a Research-Based Culture"

_socsci, doi:10.3390/socsci12060338_

Round 1
Reviewer 1 Report
•This is a solid investigation of the issue I think the authors are trying to address: how to teach methodologies courses using the actual methodologies we want our students to learn and use and to create an environment conducive to that, rather than falling into the trap of droning on and on from slides, telling them how to do things. The authors seem to eventually get to this point.
•Given that courses are for students, why didn’t the authors conduct focus groups or surveys with the students, or at least discuss why they didn’t, or didn’t include them, here.
•Is this study itself ‘scientific’? Quantitative? like the authors are arguing for in terms of teaching in their own programs? It claims it’s ‘empirical’, but I need to be convinced: the only data I see are qualitative quotations from teachers interviewed. Where is the theme analysis from their videos, for example?
•The conclusions are consistent with what we’ve known in general in education research (especially in teaching and studying teaching) for decades. Would be useful to acknowledge more of this work, at least in review, from across disciplines in teaching such courses. What does these authors’ particular study here add? Ironically, even in the education of educators, still the same issues present. . .
•Difficult among these points below to know what are simply language problems and what are deeper conceptual problems: some minor and easy to address, others confusing and unclear:
-
The paper title is tortuous and needs simplification, e.g. Teaching methods courses in Education: toward developing a research-based culture.
-
Rather than in the abstract (and throughout) saying conclusions are ‘tentative’, which makes them sound weaker than they are, perhaps say ‘proposed based on our data’, for example.
-
Whenever referring to Education departments or Education programs or courses, I recommend capitalizing Education; otherwise, it’s very confusing, because EVERY course that teaches future teachers (which are in many if not most academic departments and disciplines) is engaging in education with a small e.
-
Sentence in line 40-43: What does this mean? Untwist the jargon into shortened clearer sentences.
-
Lines 51-55. The words ‘science’ and ‘scientific’ here (and other places in ms) are confusing. Do the authors mean ‘quantitative’ or ‘hypothesis-driven’ or 'empirical" or ‘testable’ or ‘research-based’, for example? An important clarification.
-
Sentence starting on line 108 makes no sense.
-
Section 4, starting on line 190: not clear what the authors mean by ‘scenario’, language in this section in general, lines 190-202, very unclear.
-
Section 4.1, again the story is weakened by using the word ‘seems’ repeatedly. Okay just to delete this word.
-
Sentence line 316-318 very unclear wording, delete ‘seem’, state conclusion more strongly and clearly.
-
Sentence starting line 396 unclear.
-
Line 502, more torturous titles: . . . the need for . . . the need of. . .
see above
Author Response
|
Reviewer 1 |
|
|
This is a solid investigation of the issue I think the authors are trying to address: how to teach methodologies courses using the actual methodologies we want our students to learn and use and to create an environment conducive to that, rather than falling into the trap of droning on and on from slides, telling them how to do things. The authors seem to eventually get to this point.
•Given that courses are for students, why didn’t the authors conduct focus groups or surveys with the students, or at least discuss why they didn’t, or didn’t include them, here.
|
This paper is based on the ReMASE project, as described in section 2. In this exploratory project, only the teachers are considered as interlocutors - producers of knowledge called for debate.
This issue has been better described in the paper.
See lines: 74 - 81. |
|
•Is this study itself ‘scientific’? Quantitative? like the authors are arguing for in terms of teaching in their own programs? It claims it’s ‘empirical’, but I need to be convinced: the only data I see are qualitative quotations from teachers interviewed. Where is the theme analysis from their videos, for example? |
Yes, this is a scientific study. For the purpose of identifying elements of a research-based pedagogical culture, the methodological approach (for that specific objective of a larger project) was to address the key informants - the teachers - in a deep analysis of their discourse through the use of conceptual categories combined with emergent categories (see for example Cohen, Manion & Morrison, 2018, Part 3) under a pragmatic paradigm (see Creswell, 2010). Under the traditional qualitative.-quantitative distinction, the study could be tagged as qualitative although authors prefer to use Creswell framework. The empirical data analysed for this paper was the videorecording material collected in the focus-group interviews. This is largely described in section 3.
See lines: 82 – 91. |
|
•The conclusions are consistent with what we’ve known in general in education research (especially in teaching and studying teaching) for decades. Would be useful to acknowledge more of this work, at least in review, from across disciplines in teaching such courses. What does these authors’ particular study here add? Ironically, even in the education of educators, still the same issues present. . . |
We performed the changes accordingly.
See: points 5
|
|
•Difficult among these points below to know what are simply language problems and what are deeper conceptual problems: some minor and easy to address, others confusing and unclear:
. The paper title is tortuous and needs simplification, e.g. Teaching methods courses in Education: toward developing a research-based culture. |
We agree to simplify the title along the suggestion of the reviewer and changed the title into “Teaching research methods courses in Education: towards a research-based culture” |
|
. Rather than in the abstract (and throughout) saying conclusions are ‘tentative’, which makes them sound weaker than they are, perhaps say ‘proposed based on our data’, for example. |
We made the changes accordingly.
See lines: 9 – 11 / 90- 92 |
|
. Whenever referring to Education departments or Education programs or courses, I recommend capitalizing Education; otherwise, it’s very confusing, because EVERY course that teaches future teachers (which are in many if not most academic departments and disciplines) is engaging in education with a small e. |
We made the changes accordingly.
See: the document. |
|
. Sentence in line 40-43: What does this mean? Untwist the jargon into shortened clearer sentences. |
We believe that the sentence is explicit in itself, given the context and discussion that precedes it. However, we changed it in order to make it more understandable.
See lines: 40 - 44. |
|
. Lines 51-55. The words ‘science’ and ‘scientific’ here (and other places in ms) are confusing. Do the authors mean ‘quantitative’ or ‘hypothesis-driven’ or 'empirical" or ‘testable’ or ‘research-based’, for example? An important clarification. |
These lines concern to the state of the art. The words ‘science’ and ‘scientific’ cannot be described in any other way, since they refer in their amplitude to their broad meaning. There is no question of being only quantitative, as exemplified in the review. We do not consider it necessary to justify terminology that in itself has an aggregative function of information. |
|
. Sentence starting on line 108 makes no sense. |
We made the changes accordingly.
See lines: 127 - 128 |
|
. Section 4, starting on line 190: not clear what the authors mean by ‘scenario’, language in this section in general, lines 190-202, very unclear. |
We made the changes accordingly.
See: introduction, point 5.1 and references |
|
. Section 4.1, again the story is weakened by using the word ‘seems’ repeatedly. Okay just to delete this word.
. Sentence line 316-318 very unclear wording, delete ‘seem’, state conclusion more strongly and clearly.
. Sentence starting line 396 unclear.
. Line 502, more torturous titles: . . . the need for . . . the need of. . .
|
We made the changes accordingly.
See lines: 63, 71, 188, 231, 234, 287, 295, 307, 350, 376, 402, 418, 436, 444, 346, 463, 495, 527, 541 |
|
|
|

Reviewer 2 Report
The article has a clear introduction about the role of research for evidence-informed policy in education and the challenge that teaching research methods in higher education poses.
The research method and methodological approach, as well as the results are clearly presented.
The conclusion provides clear guidelines about how the research-based pedagogical culture can be applied in teaching RME courses.
Author Response
|
|
|
|
Reviewer 2 |
|
|
The article has a clear introduction about the role of research for evidence-informed policy in education and the challenge that teaching research methods in higher education poses. The research method and methodological approach, as well as the results are clearly presented. The conclusion provides clear guidelines about how the research-based pedagogical culture can be applied in teaching RME courses. |
Thank you for the comments. |
|
|
|

Reviewer 3 Report
Dear author(s), I agree with your literature review about the 'fear' of undertaking RM modules/course by undergraduates in HEI's. Some reasoning as to why masters and doctoral programs are accredited by an external agency would have helped to locate the rationalisation better.
I would like a short explanation of the conceptual map teaching/modelling in designing the M degree syllabus. After all, getting a good understanding/grounding of conceptualisation map (focus of their research using literature) is the most challenging for students (well at least those I tech/and have done for a twenty years)!
Culture of research is important- I would like you to unpack this as this is an important issue for many HEIs. How do you create this research culture when there is limited staff (time) who already have busy schedule teaching/research commitment in their own field, especially in Teacher Training programmes when staff have to visit student teachers in the field? What remain some the leadership challenges for staff management, as you need the 'buy-in' from staff, as the culture creation just doesn't materialise, does it?
Under section 3: Why isn't the HEI provider running such a course given they are the closes to the students/know the students historically and contextually (by locality/region etc)? What constituted the research team- you say 2 were involved in interviews by internet, what did other(s) do? Some notion of time taken to decipher the date should be mentioned. Are lengthy quotes necessary- why not edit with pithy observations/briefer quotes but retaining meaning, as whose I'd wonder whose voice am I really hearing in this paper- the authors or the participants?
There a hint of dissemination of students' learning, which is fine. But are the students taught about dissemination of their outcomes- 'teachers as researchers' is quite a threatening idea for teachers in the classroom- you may of course disagree.
The level of writing is good with scholarly tone to the whole article. I would sharply edit the long quotes though.
Author Response
|
|
|
|
Reviewer 3 |
|
|
Dear author(s), I agree with your literature review about the 'fear' of undertaking RM modules/course by undergraduates in HEI's. Some reasoning as to why masters and doctoral programs are accredited by an external agency would have helped to locate the rationalisation better.
|
In Portugal, among many countries, all programs in higher education need to be accredited and validated by the Agency for Assessment and Accreditation of Higher Education - A3ES, under the Decree-Law no. 369/2007 with the purpose of promoting and ensuring the quality of higher education. |
|
I would like a short explanation of the conceptual map teaching/modelling in designing the M degree syllabus. After all, getting a good understanding/grounding of conceptualisation map (focus of their research using literature) is the most challenging for students (well at least those I tech/and have done for a twenty years)! |
In accordance, the conceptual map for designing M courses in Portugal is orientated by the same agency A3ES and is available online at https://www.a3es.pt/en/about-a3es |
|
Culture of research is important- I would like you to unpack this as this is an important issue for many HEIs. How do you create this research culture when there is limited staff (time) who already have busy schedule teaching/research commitment in their own field, especially in Teacher Training programmes when staff have to visit student teachers in the field? What remain some the leadership challenges for staff management, as you need the 'buy-in' from staff, as the culture creation just doesn't materialise, does it? |
We acknowledge that the issue of limited staff time is crucial to address properly the teaching strategies, specially in teacher training programmes. The paper includes some of these issues (referred in the results. See lines: 190 – 502) but they would deserve to be scrutinized in future proper research initiatives.
|
|
Under section 3: Why isn't the HEI provider running such a course given they are the closes to the students/know the students historically and contextually (by locality/region etc)? What constituted the research team- you say 2 were involved in interviews by internet, what did other(s) do? |
The tasks that were associated with the elaboration of this paper were distributed among the authors. Each author had specific assignments. Please check the section: Author Contributions.
|
|
Some notion of time taken to decipher the date should be mentioned. |
Information about time allocated to data analysis was added in the paper. |
|
Are lengthy quotes necessary- why not edit with pithy observations/briefer quotes but retaining meaning, as whose I'd wonder whose voice am I really hearing in this paper- the authors or the participants? |
In the presentation of results we intentionally give voice to teachers,making justice to the commitment to teacher participation in the focus-group interviews. The representation of evidence implies hard choices in a short paper but we believe that we included only the relevant and pertinent quotes. |
|
There a hint of dissemination of students' learning, which is fine. But are the students taught about dissemination of their outcomes- 'teachers as researchers' is quite a threatening idea for teachers in the classroom- you may of course disagree. |
We acknowledge the point of view of the reviewer and we agree that it is quite challenging, in fact the very idea of “teacher as researcher” is rather challenging and would deserve more effort in the research on teacher education. |

Round 2
Reviewer 1 Report
The authors have addressed all this reviewer's concerns adequately. Thank you.
Looks good with minor tweaks necessary.